# SELECTIVE SEEING: CONTEXT-AWARE ATTENTION INTERVENTIONS FOR MITIGATING HALLUCINATIONS IN LARGE VISION-LANGUAGE MODELS

## ABSTRACT

Large Vision-Language Models (LVLMs) excel at multimodal tasks but are susceptible to hallucinations, generating text inconsistent with visual inputs. Existing methods mitigate hallucinations by uniformly strengthening visual signals, inadvertently amplifying irrelevant regions and spurious correlations. To address this, we present **Context-aware Attention Intervention (CAI)**, a training-free inference mechanism that embodies the idea of *"selectively seeing"*: reinforcing visual grounding only when and where it is needed. Our method first estimates token-image similarity to locate semantically relevant regions, and then conditionally amplifies their attention only for high-entropy tokens in deeper layers where visual grounding tends to degrade. This token-specific, uncertainty-aware design strengthens visual grounding without overwhelming the model with irrelevant signals. Extensive experiments show that **CAI** effectively mitigates hallucinations and achieves state-of-the-art performance across multiple benchmarks.

## 1 INTRODUCTION

Large Vision-Language Models (LVLMs) (Liu et al., 2023b; Dai et al., 2023; Bai et al., 2023; Zhu et al., 2023; Ye et al., 2023) have achieved remarkable performance on multimodal tasks such as image captioning (Li et al., 2023a), visual question answering (Liu et al., 2023b; Dai et al., 2023), and multimodal reasoning (Huang et al., 2025; Liu et al., 2025b; Zhou et al., 2025; Tan et al., 2025; Shen et al., 2025). Despite these advances, LVLMs are prone to hallucinations (Li et al., 2023b; Zhou et al., 2023; Liu et al., 2024a; Bai et al., 2024), producing outputs that are linguistically plausible yet factually incorrect or ungrounded in the visual input.

Previous work (Rohrbach et al., 2018; Li et al., 2023b; Zhou et al., 2023) often attributes hallucinations to statistical biases in large-scale training data, including frequently appearing objects and object co-occurrence. Hallucinations also stem from model-intrinsic factors, particularly the reliance on language priors from a pretrained language model (Rohrbach et al., 2018; Wu et al., 2022; Lee et al., 2023; Guan et al., 2024; Leng et al., 2024). LVLMs tend to generate outputs that are likely under the language model, even when these conflict with visual evidence, and the autoregressive decoding process can amplify early errors.

Existing strategies for mitigating hallucinations generally fall into two categories: training-based and training-free methods. Training-based approaches rely on curated datasets (Liu et al., 2023a; Yue et al., 2024; Yu et al., 2024a) for fine-tuning (Chen et al., 2023; Jiang et al., 2024; Yue et al., 2024) or reinforcement learning (Sun et al., 2023; Yu et al., 2024b; Zhao et al., 2023), aiming to alleviate hallucinations induced by statistical biases. However, the high computational cost of retraining has driven growing interest in training-free alternatives. These approaches (Leng et al., 2024; Favero et al., 2024; Liu et al., 2024c; Chen et al., 2025; An et al., 2025; Liu et al., 2025a; Zou et al., 2025; Wan et al., 2025) intervene at inference time, enriching visual information to counteract the model's tendency to over-rely on language priors.

Uniformly boosting visual attention can backfire, elevating irrelevant regions and strengthening spurious correlations that lead to hallucinations. Our analysis reveals two regularities that an effective intervention should respect. *First*, visual relevance is token-specific—different words should attend to different regions (Figure 1). *Second*, hallucination risk increases in deeper decoding layers,

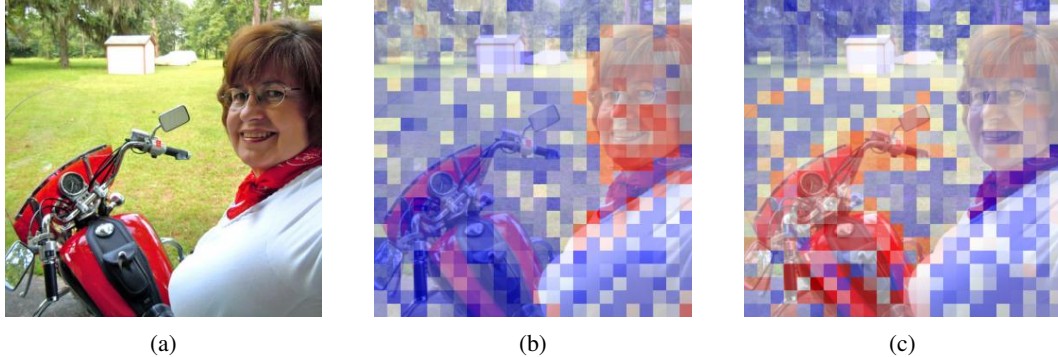

(a)                                          (b)                                          (c)

Figure 1: **Visualization of token-image similarity.** Regions highlighted in red indicate higher relevance between generated tokens and visual content. Given visual input (a) and the query *"Please describe the image in detail"*, region (b) is most associated when generating *"woman"*, whereas region (c) is most relevant when generating *"motorcycle"*.

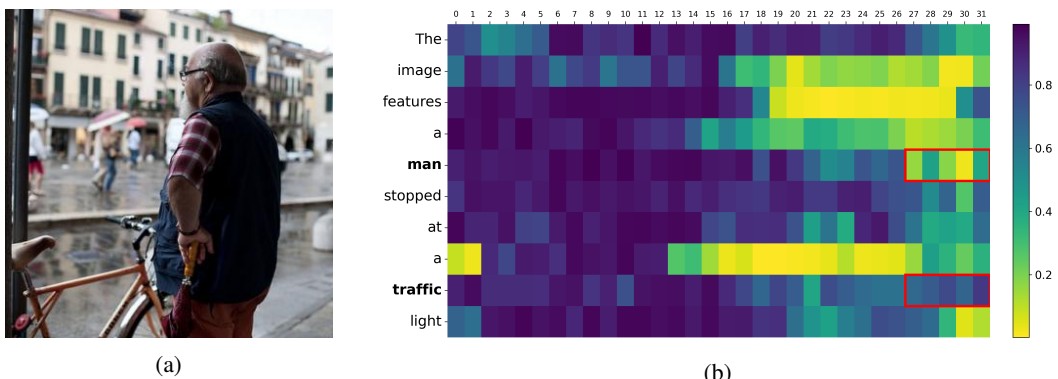

(a)                                                              (b)

Figure 2: Given visual input (a) and the query *"Please describe the image in detail"*, (b) shows **the evolution of token entropy across decoding layers**. In deeper layers, hallucination-prone tokens (e.g., *"traffic light"*) exhibit markedly higher entropy than grounded tokens (e.g., *"man"*), whereas tokens dominated by language priors (e.g., *"The", "a", "at"*) remain low-entropy.

where the predictive entropy of vulnerable tokens spikes, while function words dominated by language priors remain low-entropy (Figure 2). These observations indicate that reinforcement must be selective along two axes: (i) *where* to look—choose regions by token-image similarity; and (ii) *when* to intervene—gate by uncertainty and depth.

Building on these insights, we propose **C**ontext-aware **A**ttention **I**ntervention (**CAI**), a *training-free* mechanism that dynamically reinforces visual grounding at inference time. At each decoding step, **CAI** computes token-image similarity to select semantically relevant regions and *conditionally* amplifies attention to those regions only when the current token exhibits high predictive entropy in deeper layers, leaving low-entropy tokens and shallow layers untouched. To further counteract biases from language priors, we integrate contrastive decoding following PAI (Liu et al., 2024c), penalizing text-only hypotheses in favor of visually grounded ones. Empirical evaluations on LLaVA-1.5 (Liu et al., 2024b), InstructBLIP (Dai et al., 2023), and Qwen-VL (Bai et al., 2023) show that **CAI** consistently reduces hallucinations and outperforms prior methods on several benchmarks.

In summary, (1) We propose **CAI**, a novel training-free approach that dynamically intervenes in attention during the decoding process of LVLMs to effectively mitigate hallucinations. (2) We develop a token-specific intervention that directs attention toward the visual information most closely associated with the evolving text, avoiding interference from irrelevant regions. (3) We perform conditional interventions in deep layers under uncertainty, enabling LVLMs to achieve a balance between factual grounding and coherent text generation.

## 2 RELATED WORK

**Large Vision-Language Models (LVLMs).**   In recent LVLMs (Liu et al., 2023b; Dai et al., 2023; Bai et al., 2023; Zhu et al., 2023; Ye et al., 2023), vision-language integration enables Large Language Model (LLMs) (Brown et al., 2020; Ouyang et al., 2022; Touvron et al., 2023; Chowdhery et al., 2023; Bai et al., 2023) to extend their reasoning beyond text by incorporating visual information. Images are first processed by a vision encoder into embeddings that are aligned with the LLM's textual space (Li et al., 2019; Sun et al., 2019; Li et al., 2023a), allowing the model to interpret visual content (Li et al., 2023a), answer questions (Liu et al., 2023b; Dai et al., 2023), and generate multimodal outputs (Huang et al., 2025; Liu et al., 2025b; Zhou et al., 2025; Tan et al., 2025; Shen et al., 2025). This integration effectively empowers LLMs to perform tasks that require understanding both language and vision, bridging the gap between seeing and reasoning. Despite their capabilities, LVLMs often suffer from object hallucination (Li et al., 2023b; Zhou et al., 2023; Liu et al., 2024a; Bai et al., 2024), describing objects that are not present in the image. Such errors reduce reliability, particularly in tasks requiring precise visual understanding. Rather than changing architectures or retraining on curated data, we act *at inference time*. Our approach selectively reinforces attention to token-relevant visual evidence, improving fidelity while remaining training-free and plug-and-play across LVLM backbones.

**Mitigating hallucinations in LVLMs.**   Recent LVLM research has highlighted object hallucination as a persistent challenge. Hallucinations arise from both statistical biases in large-scale training data, such as frequently occurring objects and common object co-occurrences (Rohrbach et al., 2018; Li et al., 2023b; Zhou et al., 2023), and model-intrinsic factors, notably the reliance on language priors from pretrained language models (Rohrbach et al., 2018; Wu et al., 2022; Lee et al., 2023; Guan et al., 2024; Leng et al., 2024). LVLM outputs often align with language priors despite contradicting visual evidence. Strategies to mitigate hallucinations generally fall into training-based and training-free approaches. Training-based methods employ curated datasets (Liu et al., 2023a; Yue et al., 2024; Yu et al., 2024a) for fine-tuning (Chen et al., 2023; Jiang et al., 2024; Yue et al., 2024) or reinforcement learning (Sun et al., 2023; Yu et al., 2024b; Zhao et al., 2023) to reduce bias-induced errors, but their high computational cost limits scalability. In contrast, training-free methods (Leng et al., 2024; Favero et al., 2024; Liu et al., 2024c; Chen et al., 2025; An et al., 2025; Liu et al., 2025a; Zou et al., 2025; Wan et al., 2025) intervene at inference time, enhancing visual grounding to counteract the model's over-reliance on language priors, offering a more efficient alternative for improving output fidelity. Inspired by Neo et al. (2025), who show that visual information is localized near object tokens, we propose a *token-level*, *depth- and uncertainty-gated* method: it identifies token-image relevance and activates only when predictive entropy spikes in deeper layers. This design strengthens grounding precisely where needed and complements contrastive decoding to counter language-prior bias.

## 3 PRELIMINARY

LVLMs generalize Large Language Models (LLMs) to enable joint reasoning over textual and visual modalities. A vision encoder extracts visual features $\mathbf{v} = [v_1, \ldots, v_{N_v}]$ from an input image, while a language model encodes textual input into query tokens $\mathbf{x} = [x_1, \ldots, x_{N_x}]$. These modalities are integrated through mechanisms such as multilayer perceptrons (Liu et al., 2024b) or Q-Formers (Dai et al., 2023), generating a compact representation that conditions the language model. This fused representation facilitates autoregressive generation, formalized as:

$$y_t \sim p\left(y_t | \mathbf{v}, \mathbf{x}, \mathbf{y}_{<t}\right) = \mathcal{S}\left(f_\theta\left(y_t | \mathbf{v}, \mathbf{x}, \mathbf{y}_{<t}\right)\right). \tag{1}$$

where $y_t$ denotes the token generated at step $t$, $\mathbf{y}_{<t}$ represents the preceding token sequence, and $\mathcal{S}$ is the softmax operator over the vocabulary. Here, $f_\theta$ corresponds to the language model parameterized by $\theta$. The model is implemented as a stack of transformer blocks, each comprising multi-head self-attention (MHA) and a feed-forward network (FFN). The attention operation for head $n$ is:

$$\mathrm{Attn}_n(h) = \mathcal{S}\left(\mathbf{A}_n\right) V_n, \quad \mathbf{A}_n = \frac{Q_n K_n^\top}{\sqrt{d_k}}. \tag{2}$$

where $Q_n, K_n, V_n \in \mathbb{R}^{N \times d_k}$ are the query, key, value projections of the hidden state $h$, and $d_k$ is the key dimensionality, $N = N_x + N_v$ represents the total number of multimodal tokens. The attention

weights $\mathbf{A}_n \in \mathbb{R}^{N \times N}$ capture token-to-token dependencies, facilitating contextual feature mixing. The outputs of all $H$ heads are concatenated and projected through an output matrix $W_o$:

$$\text{MHA}(h) = \text{Concat}\left(\text{Attn}_1\left(h\right), \cdots, \text{Attn}_H\left(h\right)\right) \cdot W_o. \tag{3}$$

Finally, each transformer block applies an FFN to the MHA output, introducing nonlinear transformations that enhance contextual embeddings.

## 4 METHOD

In this section, we present Context-aware Attention Intervention (CAI), a training-free approach to mitigate hallucinations at inference time. Previous work (Leng et al., 2024; Favero et al., 2024; Liu et al., 2024c; Chen et al., 2025; An et al., 2025; Liu et al., 2025a; Wan et al., 2025) often intervenes indiscriminately across the visual input, potentially introducing noise and spurious correlations from irrelevant regions. To overcome this limitation, **CAI** exploits *token-image similarity* to quantify the semantic alignment between decoding tokens and visual regions. Guided by the similarity, **CAI** amplifies the attention weights of relevant vision tokens to reinforce visual grounding. The intervention is applied conditionally to tokens with high hallucination risk, particularly in deeper layers where visual information tends to diminish. Combined with contrastive decoding (Liu et al., 2024c), **CAI** mitigates over-reliance on language prior while reducing interference from irrelevant visual content. The pipeline of **CAI** is shown in Figure 3.

**Token-image similarity.** At step $t$, **CAI** measures the similarity between the current text token and the set of visual patch tokens:

$$\mathbf{w}_t = \sigma\left(\mathbf{v} \cdot h_t^0\right). \tag{4}$$

where $h_t^0 \in \mathbb{R}^{d_h}$ denotes the hidden state of the last token at the lowest decoder layer, where token representations are minimally entangled with higher-level abstractions, and $d_h$ is the dimensionality of the hidden state. $\mathbf{v} \in \mathbb{R}^{N_v \times d_h}$ encodes the visual features of $N_v$ image tokens. The dot product $\mathbf{v} \cdot h_t^0$ computes a similarity score for each text-patch pair, which is then normalized by $\sigma(\cdot)$ to $(0, 1)$, yielding $\mathbf{w}_t \in \mathbb{R}^{N_v}$. These weights capture fine-grained text-vision alignment and provide a stable grounding signal to guide attention in deeper layers, enhancing multimodal consistency.

---

**Algorithm 1** Context-aware Attention Intervention

**Input:** Transformer layers $L$, query embedding $\mathbf{x}$, image feature $\mathbf{v}$, image start token $i_s$, image end token $i_e$, start intervention layer $l_s$, entropy threshold $\gamma$, decoding coefficient $\lambda$.

**Output:** Response token $y_t$ at decoding step $t$.

1: **for** $l \in [0, L)$ **do**
2:     **if** $l = 0$ **then**
3:         $\mathbf{w}_t \leftarrow \sigma\left(\mathbf{v} \cdot h_t^l\right)$. % token-image similarity
4:     **end if**
5:     % conditional attention intervention
6:     **if** $l \geqslant l_s$ **and** $-\sum p\left(h_t^{l-1}\right) \log p\left(h_t^{l-1}\right) > \gamma$ **then**
7:         $\mathbf{A}_{t,i_s:i_e}^l \leftarrow \mathbf{A}_{t,i_s:i_e}^l + \left|\mathbf{A}_{t,i_s:i_e}^l\right| \odot \mathbf{w}_t$.
8:     **end if**
9:     % compute output for each layer
10:     $\bar{h}_t^l \leftarrow h_t^l + \text{MHA}_l\left(h_t^l\right)$.
11:     $h_t^{l+1} \leftarrow h_t^l + \text{FFN}_l\left(\bar{h}_t^l\right)$.
12: **end for**
13: % contrastive decoding
14: $p\left(y_t|\mathbf{v}, \mathbf{x}\right) \leftarrow \text{Linear}\left(h_t^L\right)$.
15: $\hat{p}\left(y_t|\mathbf{v}, \mathbf{x}\right) \leftarrow \lambda\, p\left(y_t|\mathbf{v}, \mathbf{x}\right) - (1 - \lambda)\, p\left(y_t|\mathbf{x}\right)$.

---

**Similarity-guided attention intervention.** Based on the similarity $\mathbf{w}_t$, **CAI** performs an attention intervention on the image regions indexed by $[i_s, i_e)$ during the generation of the $t$-th token:

$$\mathbf{A}_{t,i_s:i_e} = \mathbf{A}_{t,i_s:i_e} + \left|\mathbf{A}_{t,i_s,i_e}\right| \odot \mathbf{w}_t. \tag{5}$$

where $\mathbf{A} \in \mathbb{R}^{H \times N \times N}$ denotes the multi-head attention. The slice $\mathbf{A}_{t,i_s:i_e} \in \mathbb{R}^{H \times N_v}$, with $N_v = i_e - i_s$, represents the attention weights from the $t$-th token to $N_v$ image tokens across $H$ heads. The operator $\odot$ indicates element-wise multiplication. By scaling the attention magnitude $\left|\mathbf{A}_{t,i_s:i_e}\right|$ with the similarity $\mathbf{w}_t$, the intervention amplifies attention proportionally to semantic relevance. This mechanism reinforces context-aware visual grounding by directing the model's focus toward the image regions that are most pertinent to the token under generation.

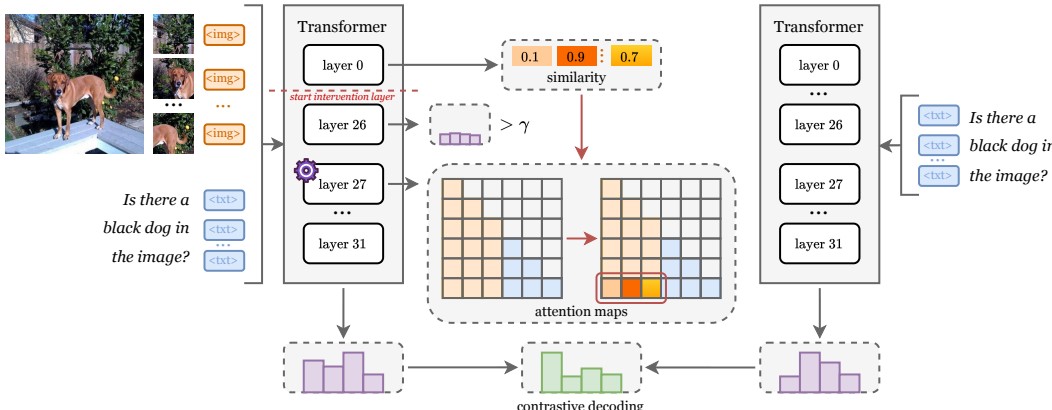

Figure 3: **Overview of CAI.** At each decoding step, the lowest layer evaluates the similarity between the generated token and each patch token (Eq. 4), yielding a visual grounding signal. Attention interventions are applied to deep decoding layers with high entropy (Eq. 6), redirecting attention toward patch tokens in proportion to their similarity (Eq. 5). Integrated with contrastive decoding (Eq.7), CAI mitigates reliance on language priors and reinforces visual grounding to reduce hallucinations.

**Conditional intervention.** Applying interventions indiscriminately across all tokens and layers risks perturbing representations that are already well-grounded. To address this, **CAI** activates interventions only under two conditions. The first confines interventions to layers deeper than $l_s$, where visual signals are susceptible to degradation. The second targets tokens with high hallucination risk, operationalized as high entropy in the hidden state:

$$-\sum p\left(h_t\right)\log p\left(h_t\right) > \gamma. \tag{6}$$

where $\gamma$ is a threshold. By conditioning on both layer depth and entropy, **CAI** focuses reinforcement on hallucination-prone tokens, ensuring that interventions are precise and effective.

**Contrastive decoding.** Following PAI (Liu et al., 2024c), hallucinations induced by over-reliance on language priors are mitigated by contrasting multimodal and unimodal predictions:

$$\hat{p}\left(y_t|\mathbf{v}, \mathbf{x}\right) = \lambda\, p\left(y_t|\mathbf{v}, \mathbf{x}\right) - (1 - \lambda)\ p\left(y_t|\mathbf{x}\right). \tag{7}$$

where $\lambda$ is a contrastive decoding coefficient. Here, $p\left(y_t|\mathbf{v}, \mathbf{x}\right)$ denotes the multimodal prediction under attention intervention, while $p\left(y_t|\mathbf{x}\right)$ represents the unimodal (text-only) prediction without intervention. The subtractive term penalizes hypotheses attributable exclusively to language priors, thereby reinforcing grounding in the visual modality.

## 5 THEORETICAL ANALYSIS

**Setup.** Let $a \in \Delta^{N-1}$ be the attention over $N$ visual tokens at the current step and $s \in \mathbb{R}^N_{\geq 0}$ the token-image similarity. When gated by $\mathbb{I}[H_t^{(l-1)} > \gamma] \cdot \mathbb{I}[\, l \geq l_0\,]$, **CAI** applies a multiplicative tilt:

$$\tilde{a}_i = \frac{a_i\,\exp(\lambda s_i)}{\sum_j a_j\,\exp(\lambda s_j)}, \qquad \lambda \geq 0,$$

else $\tilde{a} = a$. Let $x(a) = \sum_i a_i v_i$ be the aggregated visual evidence and logits $z_y = u_y + w_y^\top x(a)$, with NLL $\mathcal{L}(a) = -\log\operatorname{softmax}(z)_{y^\star}$.

**Theorem 5.1** (KL-minimality of CAI tilting). *For any baseline $a$ and similarity $s$, the distribution $q^\star(\lambda) \propto a \odot \exp(\lambda s)$ uniquely solves $\max_{q \in \Delta^{N-1}} \mathbb{E}_q[s]$ s.t. $D_{\mathrm{KL}}(q \,\|\, a) \leq \varepsilon$ for some $\lambda \geq 0$ meeting the KL budget. Thus, CAI realizes the least-change reweighting that raises expected relevance.*

**Theorem 5.2** (Entropy-gated improvement). *Assume the linearized logit model $z_y = u_y + w_y^\top x(a)$ and local smoothness of $\log\sum_y e^{z_y}$. Let $H = -\sum_y p_y \log p_y$ with $p = \operatorname{softmax}(z)$. If $H \geq$*

| | Method | Max Token = 64 | | Max Token = 128 | |
|---|---|---|---|---|---|
| | | $CHAIR_S \downarrow$ | $CHAIR_I \downarrow$ | $CHAIR_S \downarrow$ | $CHAIR_I \downarrow$ |
| **LLaVA-1.5** | Regular | 26.0 | 8.8 | 56.6 | 16.8 |
| | VCD | 23.8 | 8.2 | 59.6 | 16.6 |
| | PAI | 26.0 | 8.7 | 54.0 | 15.2 |
| | **CAI** | **17.8** | **6.9** | **39.6** | **12.4** |
| **InstructBLIP** | Regular | 29.0 | 10.2 | 54.2 | 16.7 |
| | VCD | 26.4 | 8.7 | 55.8 | **15.9** |
| | PAI | 24.6 | 8.3 | 56.8 | 16.7 |
| | **CAI** | **24.2** | **8.2** | **52.8** | 16.5 |

Table 1: **CHAIR hallucination evaluation results for LLaVA-1.5 and InstructBLIP.** The evaluation is conducted under different maximum token settings. Our approach achieves lower sentence-level ($CHAIR_S$) and instance-level ($CHAIR_I$) hallucination scores compared to the baseline.

$H_0 > 0$ *and the CAI direction aligns with the NLL descent, i.e.,* $\langle g(a), \Delta x \rangle > 0$ *where* $g(a) = \sum_y (p_y - 1\!\!1[y=y^\star]) w_y$ *and* $\Delta x = x(\tilde{a}) - x(a)$, *then there exists* $\lambda_0 > 0$ *such that for all* $0 < \lambda \leq \lambda_0$, $\mathcal{L}(\tilde{a}) < \mathcal{L}(a)$. *Hence high-entropy tokens are the regime where CAI is guaranteed to help for sufficiently small tilts.*

**Theorem 5.3** (Depth advantage via visual decay). *Suppose the visual component of hidden states evolves as* $x^{(l)} = M^{(l)} x^{(l-1)}$ *with* $\mathbb{E}\, \rho(M^{(l)}) \leq \rho < 1$. *Then for* $l \geq l_0$, $\|x^{(l)}\| \leq \rho^{l-l_0} \|x^{(l_0)}\|$. *Therefore the signal-to-noise ratio of visual evidence decays geometrically with depth, making depth-gated intervention* ($l \geq l_0$) *yield larger marginal returns.*

**Theorem 5.4** (Non-interference under small tilts). *For small* $\lambda$, $D_{\mathrm{KL}}(\tilde{a} \,\|\, a) = \frac{1}{2}\lambda^2 \mathrm{Var}_a[s] + o(\lambda^2)$; *by Pinsker,* $\|\tilde{a} - a\|_{\mathrm{TV}} \leq \sqrt{\frac{1}{2} D_{\mathrm{KL}}(\tilde{a} \,\|\, a)}$. *Thus, when the gate is off or* $s$ *is weak, CAI perturbs attention only negligibly.*

*Proofs of Theorems 5.1–5.4 are provided in Appendix B.*

# 6 EXPERIMENTS

## 6.1 EXPERIMENT SETUP

**Datasets.** Our evaluation employs three benchmarks. **POPE** (Li et al., 2023b) detects hallucinations via binary object-existence queries on MS-COCO (Lin et al., 2014), A-OKVQA (Schwenk et al., 2022), and GQA (Hudson & Manning, 2019), using *random*, *popular*, and *adversarial* sampling to assess accuracy, memorization bias, and robustness. **CHAIR** (Rohrbach et al., 2018) evaluates hallucinations in free-form captioning by quantifying both the proportion of hallucinated object instances and the proportion of captions containing hallucinations. **MME** (Yin et al., 2024) delivers a comprehensive evaluation across fourteen subtasks formulated as yes-or-no queries, encompassing perceptual dimensions such as object existence, count, position, and color.

**Implementation details.** Our evaluation is conducted on LLaVA-1.5 (Liu et al., 2024b), Instruct-BLIP (Dai et al., 2023), and Qwen-VL (Bai et al., 2023). Baseline methods, including VCD (Leng et al., 2024) and PAI (Liu et al., 2024c), are employed with their default configurations to ensure a fair comparison. For our approach, we perform a grid search over the hyperparameters $l_s$, $\gamma$ and $\lambda$, while $\sigma$ is instantiated as the sigmoid function. All experiments are executed on a single 80GB NVIDIA A800 GPU.

## 6.2 RESULTS

**Results on CHAIR.** Table 1 presents the CHAIR evaluation results for LLaVA-1.5 and Instruct-BLIP under varying maximum token settings. Compared to the baseline, **CAI** yields lower sentence-level ($CHAIR_S$) and instance-level ($CHAIR_I$) hallucination scores, indicating improved semantic fidelity and stronger visual-textual alignment across models and output lengths.

| Dataset | Method | LLaVA-1.5 | | InstructBLIP | | Qwen-VL | |
|---|---|---|---|---|---|---|---|
| | | *Accuarcy* ↑ | *F1-score* ↑ | *Accuarcy* ↑ | *F1-score* ↑ | *Accuarcy* ↑ | *F1-score* ↑ |
| MS-COCO — Random | Regular | 85.46 | 86.07 | 82.82 | 83.56 | 85.09 | 83.46 |
| | VCD | 85.74 | 86.70 | 85.53 | 85.32 | 89.59 | 89.18 |
| | PAI | 86.67 | 87.26 | 85.43 | 83.60 | 85.74 | 84.20 |
| | **CAI** | **89.24** | **89.14** | **89.55** | **89.35** | **89.90** | **89.49** |
| MS-COCO — Popular | Regular | 81.20 | 82.22 | 75.77 | 77.69 | 84.13 | 81.97 |
| | VCD | 81.93 | 83.36 | 80.97 | 81.15 | 87.57 | 87.02 |
| | PAI | 82.77 | 83.60 | 77.13 | 79.14 | 85.10 | 83.09 |
| | **CAI** | **86.03** | **85.53** | **84.30** | **83.65** | **88.30** | **87.70** |
| MS-COCO — Adversarial | Regular | 75.87 | 78.27 | 74.23 | 76.61 | 81.50 | 79.51 |
| | VCD | 76.90 | 79.62 | 79.17 | 79.99 | 84.20 | 84.07 |
| | PAI | 76.83 | 79.14 | 74.87 | 77.53 | 83.33 | 81.64 |
| | **CAI** | **82.77** | **82.73** | **81.83** | **81.56** | **84.97** | **84.69** |
| A-OKVQA — Random | Regular | 82.07 | 83.31 | 81.53 | 82.86 | 86.53 | 85.19 |
| | VCD | 82.17 | 84.14 | 85.27 | 85.72 | 89.77 | 89.50 |
| | PAI | 83.33 | 84.85 | 83.10 | 84.46 | 87.03 | 85.80 |
| | **CAI** | **89.00** | **89.03** | **89.87** | **89.73** | **90.07** | **89.91** |
| A-OKVQA — Popular | Regular | 75.30 | 78.99 | 74.80 | 77.98 | 86.00 | 84.78 |
| | VCD | 77.20 | 80.54 | 78.97 | 80.76 | 89.53 | 89.34 |
| | PAI | 76.37 | 79.79 | 76.47 | 79.61 | 87.37 | 86.17 |
| | **CAI** | **84.60** | **85.23** | **84.87** | **85.40** | **89.53** | **89.43** |
| A-OKVQA — Adversarial | Regular | 67.07 | 73.70 | 68.33 | 73.89 | 81.03 | 80.36 |
| | VCD | 68.30 | 74.86 | 73.33 | 77.00 | 82.13 | 82.92 |
| | PAI | 67.60 | 74.23 | 68.67 | 74.55 | 82.10 | 81.58 |
| | **CAI** | **77.40** | **79.79** | **75.23** | **78.04** | **82.60** | **83.20** |
| GQA — Random | Regular | 82.03 | 83.86 | 80.17 | 81.55 | 83.83 | 82.60 |
| | VCD | 81.70 | 83.99 | 83.37 | 83.78 | 88.33 | 88.26 |
| | PAI | 83.37 | 85.03 | 81.23 | 82.67 | 85.90 | 84.77 |
| | **CAI** | **89.10** | **89.15** | **88.10** | **87.84** | **90.13** | **89.87** |
| GQA — Popular | Regular | 71.93 | 76.88 | 72.27 | 75.97 | 80.77 | 80.15 |
| | VCD | 74.37 | 78.97 | 77.57 | 79.40 | 84.33 | 84.82 |
| | PAI | 72.73 | 77.60 | 73.77 | 77.34 | 82.67 | 81.89 |
| | **CAI** | **83.43** | **84.39** | **81.20** | **82.05** | **84.97** | **84.95** |
| GQA — Adversarial | Regular | 67.93 | 74.35 | 68.33 | 73.25 | 79.20 | 78.96 |
| | VCD | 68.63 | 75.41 | 73.40 | 76.38 | 81.70 | 82.74 |
| | PAI | 69.00 | 75.34 | 69.10 | 74.19 | 80.87 | 80.41 |
| | **CAI** | **78.33** | **80.46** | **76.00** | **77.65** | **83.27** | **83.51** |

Table 2: **POPE hallucination evaluation results for LLaVA-1.5, InstructBLIP, and Qwen-VL.** The evaluation is performed on the MS-COCO, A-OKVQA, and GQA datasets under different sampling strategies. Our method attains higher accuracy and F1-scores compared to the baseline.

**Results on POPE.** Table 2 demonstrates that **CAI** consistently outperforms the previous baseline in the POPE evaluation, achieving higher accuracy and F1-scores across LLaVA-1.5, InstructBLIP, and Qwen-VL. The elevated accuracy indicates effective prediction across test samples, while the higher F1-scores reflect a balanced trade-off between precision and recall, suggesting enhanced visual-textual reasoning and robust generalization.

**Results on MME.** Table 3 reports the MME evaluation, measuring object-level (existence, count) and attribute-level (position, color) reasoning. outperforms the baseline on both metrics, indicating improved visual understanding and robust reasoning.

## 6.3 DISCUSSION

**Efficiency comparison.** Table 4 summarizes the accuracy–efficiency trade-off of our two variants. The attention-only variant *CAI*[†] (i.e., **CAI** without contrastive decoding; set $\lambda=1.0$ in Eq. 7) keeps latency and throughput comparable to the baseline while delivering clear gains on POPE, CHAIR, and MME. Enabling contrastive decoding ($\lambda>1.0$) further improves performance across

| Method | Object-level | | Attribute-level | | Score ↑ |
| | Existence ↑ | Count ↑ | Position ↑ | Color ↑ | |
|---|---|---|---|---|---|
| **LLaVA-1.5** | | | | | |
| Regular | 185.00 | 126.67 | 128.33 | 148.33 | 588.33 |
| VCD | 180.00 | 141.67 | 128.33 | 153.33 | 603.33 |
| PAI | 185.00 | 131.67 | 133.33 | 153.33 | 603.33 |
| **CAI** | **190.00** | **143.33** | **148.33** | **178.33** | **660.00** |
| **InstructBLIP** | | | | | |
| Regular | 170.00 | 75.00 | 68.33 | 140.00 | 453.33 |
| VCD | 155.00 | 78.33 | 76.67 | 155.00 | 465.00 |
| PAI | 150.00 | 88.33 | **78.33** | 150.00 | 466.67 |
| **CAI** | **175.00** | **100.00** | 70.00 | **165.00** | **510.00** |
| **Qwen-VL** | | | | | |
| Regular | 165.00 | 135.00 | 163.33 | 175.00 | 638.33 |
| VCD | 170.00 | 120.00 | 133.33 | 175.00 | 598.33 |
| PAI | 175.00 | 135.00 | 163.33 | 175.00 | 648.33 |
| **CAI** | **180.00** | **146.67** | 163.33 | **185.00** | **675.00** |

Table 3: **MME evaluation results for LLaVA-1.5, InstructBLIP, and Qwen-VL.** The evaluation measures object-level reasoning, including existence and count, and attribute-level reasoning, including position and color. Our method achieves higher MME scores compared to the baseline.

| Method | Latency ↓ (ms/token) | | Throughput ↑ (token/ms) | | GPU Memory ↓ (MB) | | POPE ↑ | CHAIR ↓ | MME ↑ |
|---|---|---|---|---|---|---|---|---|---|
| Regular | 89.62 | (×1.00) | 9.41 | (×1.00) | 14241 | (×1.00) | 74.81 | 56.6 | 588.33 |
| VCD | 215.22 | (×2.40) | 2.22 | (×0.24) | 15299 | (×1.07) | 75.89 | 59.6 | 603.33 |
| PAI | 123.57 | (×1.38) | 7.09 | (×0.75) | 14281 | (×1.00) | 75.77 | 54.0 | 603.33 |
| **CAI** [†] | **92.35** | (×1.03) | **9.15** | (×0.97) | **14251** | (×1.00) | 77.13 | 43.6 | 608.33 |
| **CAI** | 131.33 | (×1.47) | 6.8 | (×0.72) | 14291 | (×1.00) | **83.67** | **39.6** | **660.00** |

Table 4: **Comparison of computational efficiency and hallucination-related performance** under LLaVA-1.5. *Latency* and *throughput* are measured per token, and *GPU memory* denotes peak usage. *POPE* represents the average accuracy on A-OKVQA across three sampling settings. *CHAIR* corresponds to $CHAIR_S$ with a maximum token length of 128. The "CAI[†]" variant sets $\lambda = 1.0$ without contrastive decoding, whereas "CAI" uses $\lambda > 1.0$ with contrastive decoding.

all three benchmarks at the cost of a modest increase in decoding time. In short: **CAI** at the attention level brings most of the benefit with near-baseline cost; adding contrastive decoding is a precision-oriented knob when hallucination risk is unacceptable.

**Effect of $l_s$ and $\gamma$ in intervention conditions.** We sweep the start layer $l_s \in [25, 30]$ and entropy threshold $\gamma \in \{0.05, 0.1, 0.15\}$ on A-OKVQA (random setting). Figure 4 peaks at $l_s = 27$ and $\gamma = 0.1$. Intervening too *early* amplifies shallow-layer noise, while intervening too *late* misses error accumulation; similarly, a threshold that is too *low* over-triggers on easy tokens, and too *high* under-triggers on genuinely uncertain tokens. These trends support our design: decide *where* to look by toke-image similarity, and *when* to act by depth and predictive entropy.

**Effect of $\lambda$ in contrastive decoding.** We grid-search the decoding coefficient $\lambda$ (Figure 5); accuracy is maximized at $\lambda = 3.0$ on A-OKVQA (random setting). Small $\lambda$ under-penalizes language-prior continuations; excessively large $\lambda$ over-restricts decoding and can hurt fluency. We use $\lambda \approx 3$ when precision is prioritized, and $\lambda = 1$ (i.e., no contrastive decoding, *CAI*[†]) in strict latency budgets.

**Case study.** Figure 6 presents a case study of MME with a visual input (a) and the query *"Is there only one zipper in the picture?"*. (b) depicts the entropy of the hallucinated response token ("Yes") generated by LLaVA-1.5, contrasted with the non-hallucinated response token ("No") produced by our **CAI**. We set the start intervention layer $l_s = 26$ and the entropy threshold $\gamma = 0.2$, observing that the prediction entropy at layers 26, 27, 29, and 30 exceeds $\gamma$, indicating a heightened risk of hallucination. Accordingly, intervention is applied at the subsequent layers, namely 27, 28, 30, and 31. During the intervention, token-image similarity (c) between the response token and input image (a) is computed at the lowest layer to establish a grounding baseline. In the high-risk layers, the atten-

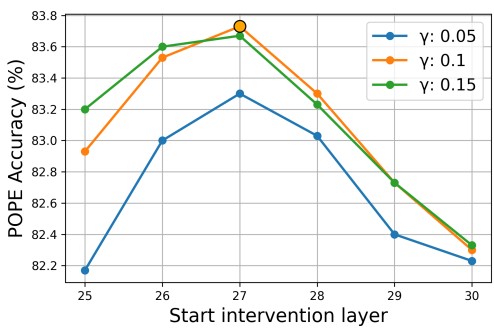

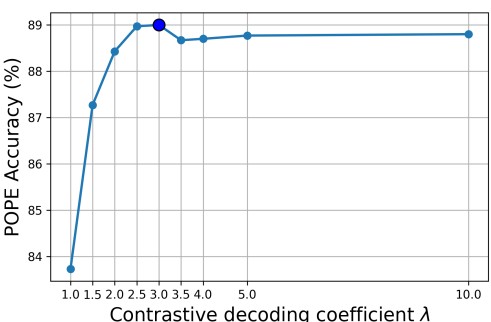

Figure 4: Ablation study on the starting intervention layer $l_s$ and the entropy threshold $\gamma$ under the random setting of A-OKVQA in POPE.

Figure 5: Ablation study on the contrastive decoding coefficient $\lambda$ under the random setting of A-OKVQA in POPE.

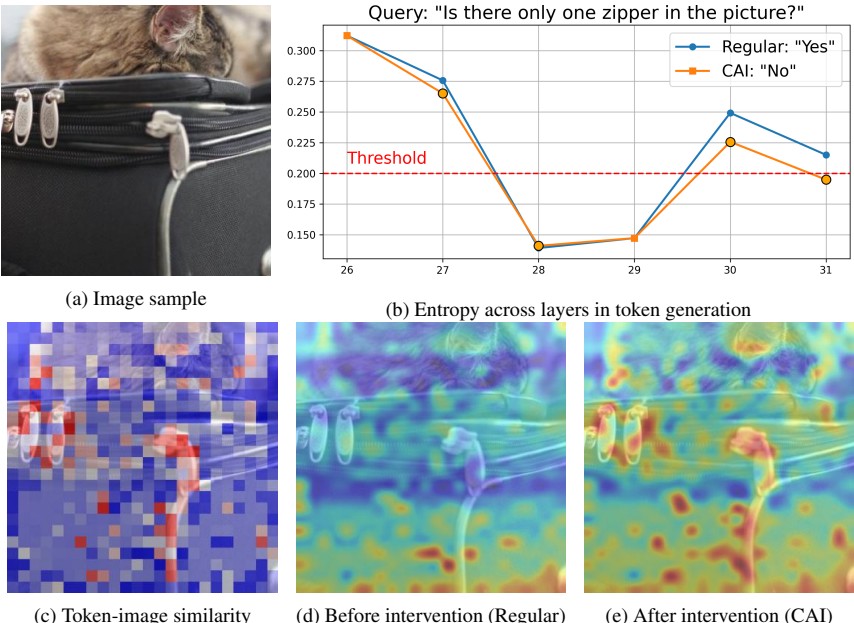

(a) Image sample

(b) Entropy across layers in token generation

(c) Token-image similarity       (d) Before intervention (Regular)       (e) After intervention (CAI)

Figure 6: **Case study** of visual input (a) with the query *"Is there only one zipper in the picture?"*. The intervention layers are shown in (b), and token-image similarity in (c) establishes grounding between the response token and the visual input. The attention maps from the $0$-th head in layer $31$, before and after the similarity-guided intervention, are visualized in (d) and (e).

tion maps visualized in (d) often lack attention to the relevant regions. By amplifying the attention weights of vision regions guided by (c), the attention map in (e) is reoriented toward token-relevant image regions and thereby mitigates hallucination. Other attention maps are shown in Figure 11.

# 7 CONCLUSION

In this work, we presented Context-aware Attention Intervention (CAI), a training-free approach to mitigate hallucinations in large vision-language models. By dynamically reinforcing attention on token-relevant visual regions and leveraging token-level uncertainty, CAI preserves fluency while suppressing irrelevant content. Combined with contrastive decoding, it improves visual grounding and mitigates language biases. Extensive evaluation demonstrates that CAI reduces hallucinations without retraining or significant computational cost, providing a practical and scalable approach for enhancing factual reliability in vision-language models.

ETHIC STATEMENT

This work introduces a training-free method to mitigate hallucinations in LVLMs. No human subjects or sensitive data were used, and all models (LLaVA-1.5, InstructBLIP, Qwen-VL) and datasets (MS-COCO, GQA, CHAIR, MME) are publicly accessible. Detailed descriptions of the methodology, hyperparameters, and evaluation protocols are provided in Sections 4, 6 and the Appendix. Source code and scripts will be made publicly available upon acceptance.

RPRODUCIBILITY STATEMENT

This work introduces a training-free method to mitigate hallucinations in LVLMs. No human subjects or sensitive data were used, and all models (LLaVA-1.5, InstructBLIP, Qwen-VL) and datasets (MS-COCO, GQA, CHAIR, MME) are publicly accessible. Detailed descriptions of the methodology, hyperparameters, and evaluation protocols are provided in Sections 4, 6 and the Appendix. Source code and scripts will be made publicly available upon acceptance.

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
