## A    LLM USAGE STATEMENT

We used a large language model (ChatGPT) solely for grammar checking and language polishing of the manuscript text. It did not contribute to research ideation, method design, experiments, data analysis, or result generation; all technical content was authored and verified by the authors.

## B    PROOFS FOR THEORETICAL RESULTS

*Proof of Theorem 5.1.* Consider $\max_{q \in \Delta^{N-1}} \mathbb{E}_q[s]$ s.t. $D_{\mathrm{KL}}(q\|a) \le \varepsilon$. The Lagrangian with multiplier $\eta \ge 0$ for the KL constraint and $\tau$ for normalization is

$$\mathcal{L}(q, \eta, \tau) = \sum_i q_i s_i - \eta \left( \sum_i q_i \log \frac{q_i}{a_i} - \varepsilon \right) + \tau \left( \sum_i q_i - 1 \right).$$

First-order optimality (KKT) gives, for each $i$, $s_i - \eta \left( \log q_i - \log a_i + 1 \right) + \tau = 0$ so that $\log q_i = \log a_i + \frac{1}{\eta} s_i - \frac{\tau + \eta}{\eta}$, hence $q_i \propto a_i \exp(\frac{1}{\eta} s_i)$. Writing $\lambda = \frac{1}{\eta} \ge 0$ yields the exponential tilt $q^\star(\lambda) \propto a \odot e^{\lambda s}$. Since $D_{\mathrm{KL}}(\cdot\|a)$ is strictly convex in $q$ and the objective is linear, the solution is unique. Moreover, with $A(\lambda) = \log \sum_i a_i e^{\lambda s_i}$, one has $D_{\mathrm{KL}}(q^\star(\lambda)\|a) = \lambda A'(\lambda) - A(\lambda)$ and $\frac{d}{d\lambda} D_{\mathrm{KL}}(q^\star(\lambda)\|a) = \lambda A''(\lambda) = \lambda \operatorname{Var}_{q^\star(\lambda)}[s] \ge 0$, so the KL budget $\varepsilon$ selects a unique $\lambda \ge 0$ by monotonicity. $\square$

*Proof of Theorem 5.2.* Let $x(a) = \sum_i a_i v_i$ and $z_y = u_y + w_y^\top x(a)$, $p = \operatorname{softmax}(z)$, and $\mathcal{L}(a) = -\log p_{y^\star}$. The gradient of $\mathcal{L}$ w.r.t. $x$ is $\nabla_x \mathcal{L}(a) = \sum_y p_y w_y - w_{y^\star} =: g(a)$. By the descent lemma for $L$-smooth functions (log-sum-exp is smooth), for $\Delta x = x(\tilde{a}) - x(a)$,

$$\mathcal{L}(\tilde{a}) - \mathcal{L}(a) \le \langle g(a), \Delta x \rangle + \frac{L}{2} \|\Delta x\|^2,$$

for some local $L > 0$ depending on $\{w_y\}$ and $p$. If the alignment condition is met (equivalently, $\langle -g(a), \Delta x \rangle > 0$, i.e., $\langle g(a), \Delta x \rangle < 0$), then the linear term is negative. Because the CAI tilt depends smoothly on $\lambda$ and $\Delta x = O(\lambda)$ for $\lambda \to 0$, there exists $\lambda_0 > 0$ such that the (negative) linear term dominates the quadratic remainder for all $0 < \lambda \le \lambda_0$, implying $\mathcal{L}(\tilde{a}) < \mathcal{L}(a)$. The entropy condition $H \ge H_0 > 0$ guarantees we are in the uncertain regime where $g(a) \ne 0$ (non-vanishing gradient), so the alignment condition is meaningful. $\square$

*Proof of Theorem 5.3.* By assumption, the visual component evolves as $x^{(l)} = M^{(l)} x^{(l-1)}$ and there exists $\rho < 1$ such that, for all $l \ge l_0 + 1$, the mixing is contractive.[1] Applying sub-multiplicativity of operator norms,

$$\|x^{(l)}\| = \|M^{(l)} M^{(l-1)} \cdots M^{(l_0+1)} x^{(l_0)}\| \le \prod_{k=l_0+1}^{l} \|M^{(k)}\| \|x^{(l_0)}\| \le \rho^{l-l_0} \|x^{(l_0)}\|.$$

Hence the visual signal decays geometrically with depth, making deeper layers yield larger marginal gains from reinforcement. $\square$

*Proof of Theorem 5.4.* For the tilted distribution $q^\star(\lambda)$ with log-normalizer $A(\lambda) = \log \sum_i a_i e^{\lambda s_i}$,

$$D_{\mathrm{KL}}(q^\star(\lambda)\|a) = \mathbb{E}_{q^\star(\lambda)} \left[ \log \frac{q^\star(\lambda)}{a} \right] = \lambda A'(\lambda) - A(\lambda).$$

A Taylor expansion of $A$ at $\lambda = 0$ gives $A(\lambda) = \mu\lambda + \frac{1}{2}\sigma^2\lambda^2 + O(\lambda^3)$ with $\mu = \mathbb{E}_a[s]$ and $\sigma^2 = \operatorname{Var}_a[s]$. Since $A'(\lambda) = \mu + \sigma^2\lambda + O(\lambda^2)$,

$$D_{\mathrm{KL}}(q^\star(\lambda)\|a) = \lambda(\mu + \sigma^2\lambda + O(\lambda^2)) - (\mu\lambda + \frac{1}{2}\sigma^2\lambda^2 + O(\lambda^3)) = \frac{1}{2}\sigma^2\lambda^2 + O(\lambda^3).$$

By Pinsker's inequality, $\|q^\star(\lambda) - a\|_{\mathrm{TV}} \le \sqrt{\frac{1}{2} D_{\mathrm{KL}}(q^\star(\lambda)\|a)} = O(\lambda)$, and any Lipschitz functional of $a$ (e.g., $w_y^\top x(a)$) is perturbed only by $O(\lambda)$. Therefore, when the gate is off or $s$ is weak (yielding small $\lambda$), the effect of CAI is negligible. $\square$

---

[1]If we further assume the spectral (operator) norm and that $M^{(l)}$ are normal, the bound $\|M^{(l)}\| \le \rho(M^{(l)}) \le \rho$ follows from $\rho(M) \le \|M\|$. In general, the same conclusion holds under the standard bounded-mixing assumption $\|M^{(l)}\| \le \rho < 1$.

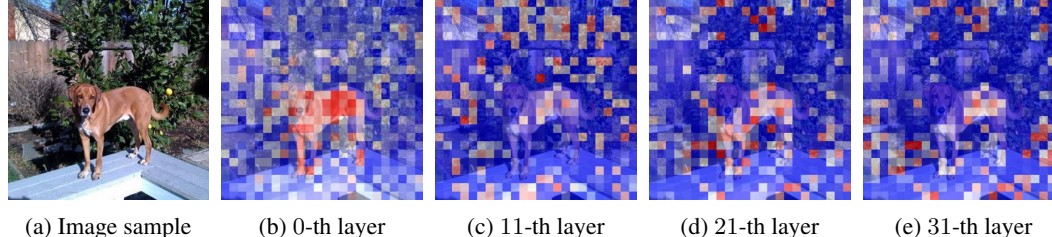

| (a) Image sample | (b) 0-th layer | (c) 11-th layer | (d) 21-th layer | (e) 31-th layer |

Figure 7: **Visualization of token-image similarity across decoding layers.** Given visual input (a) and the query *"Is there a black dog in the image?"*. It can be observed that the hidden states in the 0-th layer strongly correspond to *'dog'*, whereas the deeper layers exhibit a diminished focus on local object features, reflecting a shift toward more global representations.

## C ADDITIONAL EXPERIMENTS

### C.1 EVALUATION METRICS

**CHAIR** evaluates hallucination in image captioning by quantifying references to objects absent from the image. Using 500 randomly sampled MS-COCO images and the prompt *"Please describe this image in detail"*, we compute two complementary metrics. Sentence-level CHAIR denotes the proportion of captions containing at least one hallucinated object:

$$\text{CHAIR}_S = \frac{|\{\text{sentences with hallucinated object}\}|}{|\{\text{all sentences}\}|}, \tag{8}$$

and instance-level CHAIR is the proportion of hallucinated mentions among all object mentions:

$$\text{CHAIR}_I = \frac{|\{\text{hallucinated objects}\}|}{|\{\text{all objects mentioned}\}|}. \tag{9}$$

**POPE** serves as a binary classification task. For each image, the model is prompted with *"Is [object] in this image?"* to determine object presence. Performance is measured using standard metrics: *accuracy*, the proportion of correct predictions, and *F1-score*, the harmonic mean of precision and recall:

$$F1 = \frac{2 \times \text{precision} \times \text{recall}}{\text{precision} + \text{recall}}. \tag{10}$$

**MME** quantifies fidelity via yes-or-no questions on existence, count, position, and color. For each object, the model's responses are compared to ground truth, and the overall *MME-score* aggregates correctness across all attributes, providing a fine-grained measure of object-level hallucination and semantic alignment.

### C.2 ADDITIONAL DISCUSSIONS

**Effect of the lowest-layer representation in measuring similarity.** As shown in Figure 7, the hidden states in the 0-th layer exhibit strong correspondence to specific objects, indicating that the lowest-layer representations capture fine-grained, localized visual features. This makes them effective for measuring token-image similarity at the object level. In contrast, deeper layers shift focus toward more global and abstract representations, reducing sensitivity to individual object features.

**Effect of deep-layer entropy in detecting hallucination.** As illustrated in Figure 8, analysis of median token entropy across layers in LLaVA on the MME benchmark shows that deeper layers effectively distinguish hallucinated from non-hallucinated tokens, with hallucinated tokens exhibiting higher entropy. These findings empirically corroborate the entropy-gated improvement (Theorem 5.2) and depth advantage (Theorem 5.3), indicating that high-entropy tokens in deeper layers provide a reliable signal for identifying hallucinated outputs.

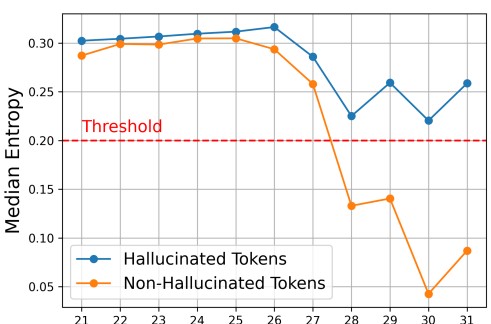

Figure 8: **Median entropy of tokens across layers.** With an intervention threshold of 0.2, hallucination and non-hallucination tokens are distinctly separable in the deeper layers.

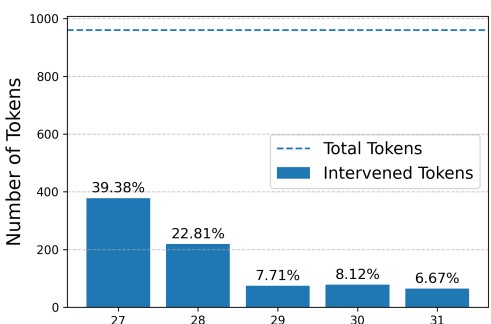

Figure 9: **Number of tokens requiring intervention at each layer**, along with their relative proportion to the total number of generated tokens.

**Efficiency of sparse token interventions.** As shown in Figure 9, in LLaVA on the MME benchmark with $l_s = 26$, **CAI** intervenes on only $6.67\% \sim 39.38\%$ of tokens in layers 27 to 31, with interventions declining in deeper layers. This empirically validates the bounded-risk property (Theorem 5.4), confirming that the method perturbs only a small subset of tokens, thereby reducing hallucinations while maintaining high inference efficiency.

**Additional case study.** Figure 10 illustrates three cases of long-text generation. The CHAIR metric is used to identify hallucinated and ground-truth tokens in "Regular" (LLaVA-1.5) responses. By applying **CAI** to reinforce visual grounding, the model amplifies the attention of response tokens toward relevant visual tokens, thereby mitigating hallucinations. Note that, due to space constraints, only the attention map of a specific head in the intervention layer is visualized here; for a more comprehensive view, Figure 11 presents the attention maps of all heads in the last layer for the MME case (Figure 6).

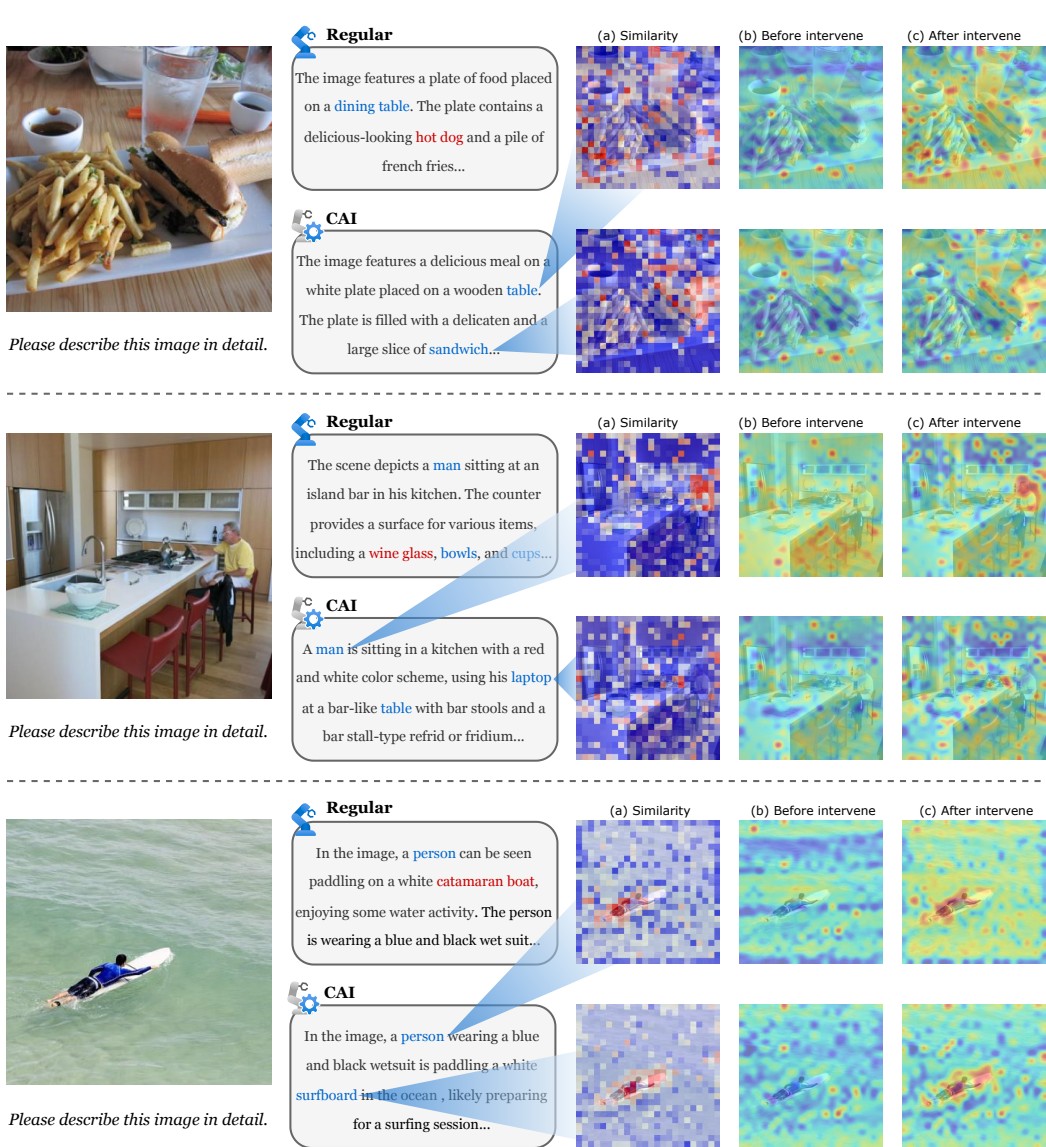

Figure 10: **Case study of CHAIR in long text generation.** Given the image and the prompt on the left, "Regular" response corresponds to LLaVA-1.5, with red denoting CHAIR hallucination words and blue representing correctly recognized terms. Employing "CAI" effectively mitigates hallucinations, with the attention intervention process illustrated in (a), (b), and (c) on the right.

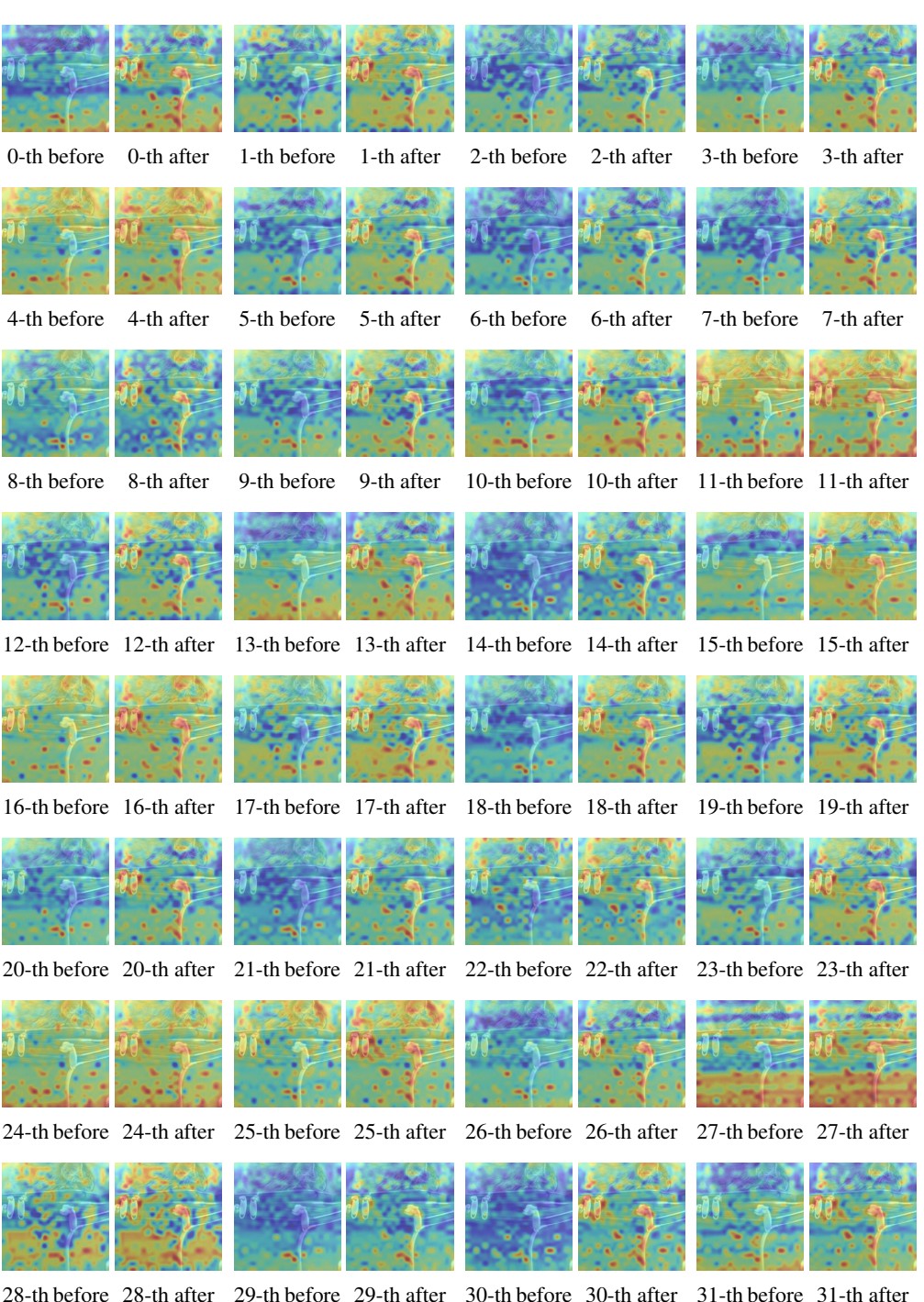

Figure 11: **Attention maps of each head in the last layer for the MME case** (Figure 6), visualized both before and after intervention.