# OpenReview forum: "Selective Seeing: Context-Aware Attention Interventions for Mitigating Hallucinations in Large Vision-Language Models"
_ICLR.cc/2026/Conference — ICLR 2026 Conference Withdrawn Submission_

### Official Review · Reviewer_hypj · 2025-10-16

**Soundness:** 2
**Presentation:** 3
**Contribution:** 2
**Rating:** 4
**Confidence:** 5

**Summary:**

This paper investigates visual attention boosting in Large Vision-Language Models (LVLMs) and identifies two empirical regularities regarding attention behaviors. Building upon these observations, it proposes Context-Aware Attention Intervention (CAI), a method that first estimates token-image similarity to locate semantically relevant visual regions, and then conditionally amplifies their attention for high-entropy tokens in deeper layers. Extensive experiments across multiple LVLMs (LLaVA-1.5, InstructBLIP, Qwen-VL) and benchmarks (POPE, CHAIR, MME) demonstrate the effectiveness of the proposed approach in mitigating visual hallucination.

**Strengths:**

- The paper is clearly written, logically organized, and supported by well-designed figures that effectively illustrate both the motivation and the method.

- The authors provide a theoretical analysis that formally justifies the design of the intervention mechanism, enhancing the paper’s credibility and interpretability.

- Experiments on multiple LVLMs and benchmarks consistently demonstrate the effectiveness of CAI in reducing hallucinations.

**Weaknesses:**

- **Limited baselines**: The paper only compares CAI with two relatively early baselines, VCD and PAI. Several recent hallucination mitigation methods are omitted, such as SID [1], VTI [2], VASparse [3], and CMI-VLD [4]. These works respectively explore vision-token preservation, latent-space steering, vision-aware decoding, and adaptive cross-modal consistency. It would strengthen the paper to include comparisons with these approaches.
- **Experimental Models**: The evaluation is conducted on a select set of mid-scale LVLMs (e.g., LLaVA-1.5, InstructBLIP), while more capable and recent models such as Qwen2.5-VL, LLaVA-NEXT and InternVL are not considered.
- **Benchmark coverage**: Incorporating broader and more challenging benchmarks, such as MMBench [5] or the GPT-4-assisted hallucination benchmark [6], would improve the comprehensiveness of the analysis.
- **Hyperparameter sensitivity**: CAI involves several hyperparameters (e.g., entropy threshold $\gamma$, intervention depth $\lambda$) and requires model-specific tuning. This dependence may limit generalizability and increase the cost of deployment across different LVLMs.
- **Missing citation**: Figure 1 and its motivational setup are conceptually similar to the recently proposed CMI-VLD [4], yet the paper does not cite or discuss this connection. Proper acknowledgment and differentiation are needed.

[1]Huo Fushuo, et al. Self-introspective decoding: Alleviating hallucinations for large vision-language models. In ICLR 2025.

[2]Liu Sheng, et al. Reducing hallucinations in vision-language models via latent space steering. In ICLR 2025

[3]Zhuang Xianwei, et al. Vasparse: Towards efficient visual hallucination mitigation for large vision-language model via visual-aware sparsification. In CVPR 2025

[4]Fang Hao, et al. Grounding Language with Vision: A Conditional Mutual Information Calibrated Decoding Strategy for Reducing Hallucinations in LVLMs. In NIPS 2025.

[5]Liu Yuan, et al. Mmbench: Is your multi-modal model an all-around player?. In ECCV 2024.

[6]Zhao, Zhiyuan, et al. Beyond hallucinations: Enhancing lvlms through hallucination-aware direct preference optimization. arXiv preprint arXiv:2311.16839 (2023).

**Questions:**

- Why is Qwen-VL not included in the CHAIR hallucination benchmark?

- Is CAI’s effect restricted to short generations? As is well known, hallucination often increases with longer outputs. Given that POPE and MME involve short yes/no responses, and CHAIR is evaluated with maximum lengths of 64 or 128 tokens (as opposed to the 512-token setting in mainstream works [1-3]), it would be important to assess CAI’s robustness in long-form generation scenarios.

- In Figure 4, the best performance occurs at  $\gamma$ = 0.1, whereas Figures 6 (b) and Figure 8 use  $\gamma$ = 0.2.

---

### Official Review · Reviewer_VXbB · 2025-10-20

**Soundness:** 2
**Presentation:** 3
**Contribution:** 2
**Rating:** 2
**Confidence:** 4

**Summary:**

This paper presents two main contributions on their method CAI: **1)** It measures the similarity between image and text tokens in **shallow layers** to amplify attention on relevant regions, and **2)** When entropy is high in **deeper layers**, CAI strengthens the visual prior using contrastive decoding to reduce over-reliance on the language prior. With these two techniques, the method achieves high accuracy and F1 scores on VLM tasks that require captioning specific objects.

**Strengths:**

Under the assumption of simple VLM tasks that do not require complex reasoning, the idea of extracting image-text relevance in shallow layers and augmenting this information is a strong concept. Moreover, analyzing the language prior based on entropy in deeper layers represents a more advanced perspective compared to previous ideas.

**Weaknesses:**

1. **Weak Analysis of Intervention Layer (very major)**: The analysis of the intervention layer, which determines the crucial distinction between "shallow" and "deeper" layers, is notably weak. The paper currently justifies its choice by showing an accuracy curve that varies with the start layer for a specific task. This approach could be considered using *"cheat-sheet" information*, as the optimal intervention point will likely vary across different models and tasks. A more practical method, perhaps a proxy, is needed to determine this point without having to run the entire task to find an empirical optimum. Without such a method, it cannot be considered a practical methodology.

2. **Concern on Linguistic Fluency Degradation (major)**: Using contrastive decoding inherently suppresses the language prior, leading to an over-reliance on the visual prior. This class of methods can degrade perplexity and linguistic naturalness, a fact that has been verified in several other papers [1,2,3,4]. Consequently, an evaluation is needed to verify whether CAI can avoid harming linguistic fluency.

3. **Limited Applicability to Complex Tasks (major to minor)**: If image relevance is assessed only in shallow layers, the model will likely struggle to extract the logically complex relationships required for higher-level VLM tasks. While this issue seems minor for the simple captioning tasks presented in this paper, the selective attention mechanism could become a disadvantage as task complexity increases. Additionally, because the methodology forcibly suppresses the language prior, it risks compromising the VLM's logical reasoning capabilities. This raises concerns that the methodology may be limited to simple tasks.

4. **Simplistic Baselines (major to minor)**: The baselines are too simplistic. Since the paper employs a contrastive decoding approach, more extensive comparisons with related methods are necessary. VCD, published in 2023, has been widely cited and has inspired numerous improved techniques. At least one or two state-of-the-art baselines should have been included for comparison. Furthermore, including a baseline like ClearSight or SumGD [3,4], which preserves the language prior and offers an alternative to contrastive decoding, would have greatly enriched the paper's analysis.

[1] GECOR: A Greedy-based Contrastive Decoding Strategy for Faithful and Coherent Text Generation

[2] Cross-Image Contrastive Decoding

[3] ClearSight: Visual Signal Enhancement for Object Hallucination Mitigation in Multimodal Large language Models

[4] Mitigating Hallucinations in Large Vision-Language Models via Summary-Guided Decoding

**Questions:**

Please see weaknesses.

---

### Official Review · Reviewer_7nDF · 2025-10-31

**Soundness:** 3
**Presentation:** 3
**Contribution:** 3
**Rating:** 4
**Confidence:** 4

**Summary:**

The paper proposes CAI, a training‑free method that selectively reinforces attention to token‑relevant image regions, gated by entropy (uncertainty) and layer depth. It shows consistent gains on CHAIR, POPE, and MME across LLaVA‑1.5, InstructBLIP, and Qwen‑VL. The method is simple to implement, adds modest overhead, and is accompanied by supportive theory (KL‑minimal exponential tilting; depth‑decay advantage; small‑tilt safety) and visual diagnostics.

**Strengths:**

1. Well‑motivated selective intervention. Empirical and theoretical support for acting only at deeper layers and under high uncertainty; lowest‑layer similarity provides a stable, object‑level grounding cue.

2. Clear performance gains across tasks/backbones. Substantial improvements on CHAIR/POPE/MME

3. Attention‑only CAI is near‑baseline latency/memory; adding contrastive decoding trades a bit of speed for extra accuracy

4. Visualizations and token‑level intervention statistics align with the method’s design and help interpret behavior.

5. Plug‑and‑play across LVLMs. Results on three diverse backbones suggest decent generality

**Weaknesses:**

1. Missing recent training-free baselines. Despite citing newer methods (e.g. ONLY one-layer intervention; latent-space steering), they are absent from experiments. Please add them to comparison or specify why can't compare with them.

2. Contrastive decoding formulation appears inconsistent. Eq. 7 uses probabilities and, with λ > 1, turns the subtractive term into addition, contradicting the stated “penalize text‑only hypotheses” goal. This is either a notational error or an implementation mismatch.

3. The intervention criteria is not clear (Eq. 6). The intervention is triggered by the intermediate layer hidden state h_t. How hidden state h_t is not a probability. How to convert h_t into a probability? The LM head?

**Questions:**

1. Could you please add more newer baselines?

2. Please further check and clarify Eq. 7.

3. Please explain more details on how to convert hidden states into probability for entropy calculation.

---

### Official Review · Reviewer_h4Kp · 2025-11-04

**Soundness:** 2
**Presentation:** 3
**Contribution:** 3
**Rating:** 4
**Confidence:** 3

**Summary:**

This paper proposes Context-aware Attention Intervention (CAI), a training-free, inference-time method to mitigate hallucinations in Large Vision-Language Models. Grounded in the principle of "selective seeing," CAI dynamically reinforces visual grounding only when and where it is needed by first computing token-image similarity to locate relevant visual regions, and then conditionally amplifying attention to these areas exclusively for high-entropy (uncertain) tokens in deeper layers. This targeted intervention, combined with contrastive decoding, effectively strengthens visual grounding and reduces reliance on faulty language priors

**Strengths:**

1. The paper presents two well-defined and well-justified motivations, and Figures 1 and 2 provide convincing support for these ideas. Unlike methods that indiscriminately amplify visual signals, CAI's "selective seeing" approach is highly targeted.
2. The experimental results in the paper show almost consistent performance improvements, demonstrating the effectiveness of the proposed method.
3. The writing in the paper is coherent and easy to understand.

**Weaknesses:**

1. The practical utility of this work is questionable, as it is only validated on outdated LVLMs. Given the rapid advances in the field, it is crucial to demonstrate its effectiveness on more recent, state-of-the-art LVLMs.
2. While CAI effectively reduces hallucinations, its potential side effects on the overall quality of text generation need to be evaluated. The authors should provide quantitative results from standard text generation tasks to address this concern. A slight performance drop is acceptable, but a severe decline may outweigh the benefits.
3. The evaluation primarily relies on benchmarks that focus on object-level hallucinations (e.g., object existence, count, color). However, hallucinations can occur in more diverse and subtle forms beyond object presence, such as inaccurately describing actions or events (e.g., stating “a man is running” when he is actually standing still). This limitation may undermine the impact of the proposed method.

**Questions:**

See the Weaknesses.

---

### Note · Authors · 2025-11-13

I have read and agree with the venue's withdrawal policy on behalf of myself and my co-authors.